# Oncostatin M-Enriched Small Extracellular Vesicles Derived from Mesenchymal Stem Cells Prevent Isoproterenol-Induced Fibrosis and Enhance Angiogenesis

**DOI:** 10.3390/ijms24076467

**Published:** 2023-03-30

**Authors:** Sandra Tejedor, Marc Buigues, Hernán González-King, Andreia M. Silva, Nahuel Aquiles García, Niek Dekker, Pilar Sepúlveda

**Affiliations:** 1Regenerative Medicine and Heart Transplantation Unit, Instituto de Investigación Sanitaria La Fe, 46026 Valencia, Spain; 2Discovery Biology, Discovery Sciences, BioPharmaceuticals R&D, AstraZeneca, 43183 Gothenburg, Sweden; 3Early Cardiovascular Renal and Metabolism (CVRM), Bioscience Cardiovascular, AstraZeneca, 43183 Gothenburg, Sweden; 4GECORP, Av. Juan Manuel de Rosas, Buenos Aires 7220, Argentina; 5Centro de Investigación Biomédica en Red Enfermedades Cardiovasculares (CIBERCV), Carlos III Institute of Health, 28029 Madrid, Spain

**Keywords:** extracellular vesicles, Oncostatin M, fibrosis, isoproterenol, mesenchymal stem cells

## Abstract

Myocardial fibrosis is a pathological hallmark of cardiac dysfunction. Oncostatin M (OSM) is a pleiotropic cytokine that can promote fibrosis in different organs after sustained exposure. However, OSM released by macrophages during cardiac fibrosis suppresses cardiac fibroblast activation by modulating transforming growth factor beta 1 (TGF-β1) expression and extracellular matrix deposition. Small extracellular vesicles (SEVs) from mesenchymal stromal cells (MSCs) are being investigated to treat myocardial infarction, using different strategies to bolster their therapeutic ability. Here, we generated TERT-immortalized human MSC cell lines (MSC-T) engineered to overexpress two forms of cleavage-resistant OSM fused to CD81TM (OSM-SEVs), which allows the display of the cytokine at the surface of secreted SEVs. The therapeutic potential of OSM-SEVs was assessed in vitro using human cardiac ventricular fibroblasts (HCF-Vs) activated by TGF-β1. Compared with control SEVs, OSM-loaded SEVs reduced proliferation in HCF-V and blunted telo-collagen expression. When injected intraperitoneally into mice treated with isoproterenol, OSM-loaded SEVs reduced fibrosis, prevented cardiac hypertrophy, and increased angiogenesis. Overall, we demonstrate that the enrichment of functional OSM on the surface of MSC-T-SEVs increases their potency in terms of anti-fibrotic and pro-angiogenic properties, which opens new perspectives for this novel biological product in cell-free-based therapies.

## 1. Introduction

Interstitial myocardial fibrosis is a common pathological event in various cardiovascular diseases that is associated with excessive extracellular matrix (ECM) deposition, which induces morphological and functional alterations in the myocardial architecture. Activation of heart resident fibroblasts in response to stress/injury induces a myofibroblast phenotype, which promotes fibrosis [1]. Interestingly, tissue-resident macrophages (Mφ) have been shown to prevent this activation [2], and the contribution of both fibroblasts and Mφ populations to cardiac healing is dynamic. While ECM deposition by activated fibroblasts is a protective mechanism that can be beneficial for wound healing and tissue regeneration in the acute phase after heart damage, chronic activation of fibrosis leads to tissue remodeling and heart failure (HF) [3]. Accordingly, new therapeutic approaches to control fibroblast activation and interstitial fibrosis are a clinical aspiration [4].

Small extracellular vesicles (SEVs) secreted by mesenchymal stromal cells (MSCs) have recently emerged as a promising tool to improve cardiac function by delivering molecular cargo to modulate apoptosis, reactive oxygen species production, angiogenesis, fibrosis, and inflammation [5,6,7,8,9,10,11]. In support of this concept, SEVs found in the secretome of human cardiac stem cells (CSCs) had the ability to reduce fibrotic scar tissue formation, interstitial fibrosis, and cardiomyocyte hypertrophy when encapsulated and implanted subcutaneously into infarcted rats [12]. SEVs have several advantages as “drug delivery” agents, including an intrinsic similarity to cell membranes, a unique biomolecule profile, a deformable structure [13], a negative zeta potential for long-term circulation [14,15], biocompatibility [16], a small size for penetration into deep tissues [17], and the possibility to be modified to augment their therapeutic activity. Accordingly, they have garnered much research interest for cell-free-based therapies for several clinical applications, including cardiac repair. To date, however, only one clinical trial using SEVs is actively recruiting patients with acute myocardial infarction (AMI) [18].

Oncostatin M (OSM), a member of the interleukin-6 cytokine family, is a pleiotropic cytokine secreted by T cells, monocytes, Mφ, dendritic cells, and neutrophils in response to hypoxia, and is regulated by hypoxia inducible factor-1α [19,20,21]. At the structural level, human OSM is initially synthesized as a 252-amino-acid polypeptide organized into three domains: a signal peptide (SP), a main chain, and a propeptide (PP). After translation, the SP and PP domains are excised by proteolytic cleavage inside of cells producing mature OSM, which is then secreted into the extracellular space, where it interacts with two heterodimeric receptor complexes: Type I and Type II. The Type I receptor complex is formed by the OSM receptor (IL-31RB) and the leukemia inhibitory factor receptor (LIF-R), also known as glycoprotein 130 (gp130), whereas the Type II receptor complex consists of IL-31RB and IL-31RA [22,23,24]. The role of OSM and its receptor complexes in the context of cardioprotection, regeneration, and failure has been recently reviewed in detail [25]. The signaling pathways and functional effects triggered by the interactions between OSM and its receptors are not yet fully understood and vary between cell types and physiological states, but these interactions induce the activation of Janus Kinase (JAK) family members (JAK1-2 and TYK2) through tyrosine phosphorylation, which in turn induces the activation and nuclear translocation of STAT proteins (STAT1, 3, 5A, and B) in receptor cells. In relation to the cardiovascular system, OSM has been linked to proliferation, chemotaxis, and endothelial tube formation in human microvascular cells [26,27,28], and its delivery using recombinant proteins or its natural delivery by restorative Mφ can activate myogenesis and angiogenesis in vitro and enhance revascularization and tissue regeneration in a murine model of acute skeletal muscle injury [29]. OSM has also been reported to promote cardiomyocyte dedifferentiation and cardiac tissue recovery after cardiac injury [30,31]. Strikingly, although OSM is considered to be a proinflammatory cytokine that promotes fibrosis in different tissues, including liver and lung tissue [32,33], during ischemic cardiac remodeling, OSM secreted by Ly6Chi monocytes/macrophages suppresses cardiac fibroblast activation by inhibiting transforming growth factor beta 1 (TGF-β1) [19] and ECM deposition [25,34]. Moreover, in a model of myocardial infarction, intraperitoneal injection of recombinant OSM (rOSM) improved cardiac function at least in part by promoting the shift of cardiac Mφ from M1 to M2 phenotype [35] and treatment of mice subjected to myocardial infarction with rOSM-alleviated post-infarction cardiac remodeling through the enhancement of cardiomyocyte autophagy [36].

In the present study, we wanted to take advantage of the potential capacity of SEVs to cross biological barriers and to efficiently participate in cellular communication processes [37]. We used a human TERT-immortalized MSC line (MSC-T) as a source of SEVs. The cell line was genetically modified to secrete a cleavage-resistant variant of OSM loaded onto the surface of SEVs through its fusion to CD81TM. The later construct includes CD81 and the transmembrane helix of PDGFRβ fused to CD81 N-terminus, what ensures the extraluminal expression of CD81 on SEVs surface [38]. We show that SEVs derived from MSC-T engineered to express OSM on its surface have an enhanced anti-fibrotic potential in in vitro and in vivo models of cardiac fibrosis.

## 2. Results

### 2.1. Generation of MSC-T Cells Overexpressing a Modified Version of OSM on the Membrane of Secreted SEVs

We started by validating our engineering approach by using Expi293F cells to load native OSM and CD81TM on SEVs. The Expi293F transfection system provides an enhanced cell-growing and EV-secretion rate compared to MSC [39]. As shown in Appendix A, OSM protein was not detected in SEVs or the correspondent cell lysates of Expi293F at the expected molecular weight for the fusion protein (OSM-TM-CD81, ~47 kDa) when a lentiviral plasmid to load native OSM into SEVs was used. However, endogenous expression of OSM was detected in all three experimental groups of cell lysates (control, CD81TM, and OSM-CD81TM) at its native molecular weight (25 kDa). The fact that OSM expression in Expi293F was not detected at its expected recombinant molecular weight was indicative of the cleavage of OSM from CD81TM under physiological conditions.

OSM is enzymatically processed for secretion as a mature protein (main chain) into the extracellular space by producer cells. Processing occurs at two cleavage sites (CSs): one between the SP and the main chain and the other between the main chain and the PP (Figure 1a). To overcome the cleavage issue, and to achieve the correct loading of the OSM onto SEVs, we designed two mutated versions of OSM that included a specific sequence modification to substitute an arginine (R) for a glycine (G) in CS2 (Figure 1a, middle and right positions), which abolishes its recognition by proteases. This mutation is located between the main chain and the PP. The first variant contained the OSM sequence lacking the PP sequence (matOSM, shown in the middle), and the second variant included the full sequence of OSM (mutOSM, shown in the right). In all versions, OSM sequences were fused to CD81TM.

Thereafter, MSC-T cells were transfected with the newly designed plasmids. The first engineered cell line contained the plasmid to load the matOSM form, and the second contained the plasmid to load the mutOSM form, both connected to CD81TM for EV surface display. The third one contained only CD81TM and was used as control SEVs. In all cases, OSM expression was regulated by an inducible promotor (Tet-on), since constitutive expression or exposure of MSC-T to recombinant OSM (rhOSM) reduced the cell-proliferation and migration rate (Appendix A) [19].

Following lentiviral transduction, genetically modified MSC-T variants (which, from now on, are referred to as CD81, matOSM-CD81, and mutOSM-CD81) were expanded, and secreted SEVs were isolated and characterized. Non-transfected MSC-T cells and SEVs were included as an additional control. A schematic representation of the experimental process is shown in Figure 1b (Step 1). The success of transduction was evaluated by measuring the expression of GFP (as a separate transcription unit controlled by a constitutive promoter contained in all plasmids), using flow cytometry (Appendix A). More than 87% of the MSC-T cells transduced with the different plasmids were positive for GFP, and the percentage of GFP-positive cells was maintained above 85% in the three cell lines generated during cell passages for experiments (Appendix A). The evaluation of an MSC marker profile established by the Mesenchymal and Tissue Stem Cell Committee [40] showed that MSC-T transduction did not change the identity of MSC cells (Appendix A). For expansion and SEV isolation, MSC-T, MSC-T-CD81, MSC-T-matOSM-CD81, and MSC-T-mutOSM-CD81 cells were incubated in SEV recollection medium (RM) containing 10 µg/mL doxycycline for 48 h (Figure 1b, Step 2). Collected media from cell cultures were used for SEV isolation by differential ultracentrifugation and filtration (Figure 1b, Step 3), and cells were used to obtain protein samples for analysis. Subsequently, the presence of recombinant proteins in the generated cell lines and SEVs isolated from supernatants was evaluated. The representative Western blotting results for OSM detection in SEVs and cell protein samples are shown in Figure 2a. The results confirmed that matOSM-CD81 and mutOSM-CD81 recombinant proteins were present in both cell extracts and SEVs, where a band of ~45–47 kDa was observed, but not in control MSC-T and CD81-SEVs and cell lysates.

SEVs isolated from the RM of MSC-T, MSC-T-CD81, MSC-T-matOSM-CD81, and MSC-T-mutOSM-CD81 cultures were further characterized by electron microscopy and nanoparticle tracking analysis (NTA). Electron microscopy showed that the four SEV groups exhibited the expected convex morphology (Figure 2b), and the NTA analysis revealed a similar size distribution in all groups, with most particles being 90 to 150 nm (Figure 2c,d). Moreover, the concentration of particles isolated from the same amount of RM after seeding the same number of cells in flasks from each condition was similar in all groups, indicating that genetic modifications did not influence the secretion of SEVs (control-SEVs, 7.62 × 10^8^ ± 4.8 × 10^7^ particles/mL; CD81-SEVs, 9.71 × 10^8^ ± 4.93 × 10^7^ particles/mL; matOSM-CD81-SEVs, 1.12 × 10^9^ ± 1.26 × 10^8^ particles/mL; and mutOSM-CD81-SEVs, 9.18 × 10^8^ ± 1.08 × 10^8^ particles/mL).

The isolated SEVs were also analyzed using the ExoView platform (ExoView^®^ R100; NanoView Biosciences, Boston, MA, USA) for immunophenotyping of different SEVs subpopulations at the single-vesicle level (see Section 4). SEVs were captured and immobilized on the silicon substrate array functionalized with an antibody for SEV capture. Spots functionalized with an IgG isotype were used as controls of non-specific SEV adsorption to the chip substrate, with no significant numbers of particles counted. SEVs from the three experimental groups were captured with CD63, and sequestrated SEVs were co-stained with antibodies against OSM, CD81, and tetraspanin-CD9 to visualize the pattern of SEV populations. Additional experiments were performed with arrays functionalized with CD9 and CD81 (Appendix A, respectively). The number of CD63/CD81 and CD63/CD9 double-positive SEVs was unchanged among the four experimental groups (Figure 2e). However, the expression of CD81 stimulated the release of triple positive CD63/CD81/CD9 SEVs by parental cells (Figure 2f). As expected, however, a small proportion of SEVs expressed OSM only in the MSC-T-matOSM-CD81 and MSC-T-mutOSM-CD81 experimental groups (Figure 2e,f).

The proportion of SEVs that were double positive for CD63 and OSM was similar in matOSM-CD81- and mutOSM-CD81-SEVs, indicating that the PP did not influence the distribution of the recombinant protein on the SEV surface. Nonetheless, the presence of PP influenced the levels of protein loaded per vesicle, since matOSM-CD81-SEVs carried much more recombinant protein than mutOSM-CD81-SEVs, as assessed by the Western blot (Figure 2a). When SEVs were captured using CD9 or CD81, we also observed the presence of OSM in matOSM-CD81-SEVs and mutOSM-CD81-SEVs (Appendix A).

Overall, the results establish the feasibility of anchoring a typically soluble cytokine (OSM) onto SEVs secreted by MSC-T by modifying a unique amino acid in its sequence. Importantly, genetically modified cells and SEVs showed no changes to their properties after introducing the new recombinant proteins.

### 2.2. OSM Receptors Are Upregulated in Human Cardiac Ventricular Fibroblasts upon In Vitro Starvation and TGFβ-1 Stimulation

The potential of OSM to counteract the pro-fibrotic effects of TGFβ-1 in cardiac ischemia models has been reported previously [19,41]. To exert its function, OSM needs to be secreted into the extracellular space and interact with either the Type I Receptor (heterodimer composed by IL-31RA and LIF-R/GP130) or the Type II receptor (IL-31RA and IL-31RB) on target cells. Thus, we next evaluated the protein levels of the three OSM receptors in human cardiac ventricular fibroblasts (HCF-Vs) after oxygen and glucose deprivation (starved) and after in vitro stimulation with a pro-fibrotic cocktail comprising L-ascorbic acid 2-phosphate, dextran sulphate and recombinant TGFβ-1. Non-treated HCF-V cells were used as a control. Western blotting (Figure 3a–d) revealed that the IL-31RA receptor levels were comparable between the control and starved cells, they but increased significantly 2 h after stimulation with the pro-fibrotic cocktail, pointing to a rapid production of this protein under these experimental conditions (*p* < 0.05 after 2 h of stimulation). GP130 protein expression was higher in starved cells than in control cells and increased from 0.5 to 4 h of stimulation (*p* < 0.001 and *p* < 0.01, respectively). Finally, in contrast to IL-31RA, the levels of IL-31RB were lower in stimulated HCF-V cells across all time points than in control and starved cells (*p* < 0.001 at different time points of stimulation, except after 1 h of stimulation, which was *p* < 0.01).

In summary, the results indicate that receptors involved in OSM signaling pathway activation are upregulated in starved HCF-V cells and at early time points after in vitro fibrosis stimulation, pointing to their potential responsiveness to OSM loaded into SEVs.

### 2.3. MutOSM-CD81-Enriched SEVs Suppress the Proliferation of Human Cardiac Ventricular Fibroblasts after Starvation

We then assessed the potential impact of OSM-enriched SEVs on HCF-V proliferation after starvation, using Ki-67 nuclei immunostaining. As shown in Figure 4a,b, proliferation was higher in starved cells than in control cells. Moreover, proliferation rates were similar between untreated starved control cells and cells treated with CD81-SEVs. A slight decrease in proliferation was observed in HCF-V cells treated with matOSM-CD81-SEVs when compared to untreated cells or cells treated with control SEVs, although this was not significant. By contrast, treatment of HCF-V cells with mutOSM-CD81-SEVs significantly decreased their proliferation in comparison with starved cells or cells treated with control SEVs.

Overall, these results point to a functional role for mutOSM-SEVs in modifying HCF-V proliferation after starvation in vitro, indicating that loading a modified version of OSM onto the surface of MSC-T-SEVs might promote anti-fibrotic responses in ventricular fibroblasts.

### 2.4. MatOSM- and mutOSM-Enriched SEVs Counteract Telo-Collagen1α1 Expression in an In Vitro Model of Cardiac Fibrosis

Given the above results, we next monitored the fibrotic phenotype of HCF-V cells starved and stimulated for fibrosis induction in response to SEVs by measuring the expression of mature collagen via immunocytochemistry. The telo-Collagen1α1 fluorescent signal was quantified by two different methods: measuring mean fluorescence intensity (MFI) per number of cells (nuclei stained in blue using DAPI, Figure 4c), and by the area occupied by extracellular telo-Collagen1α1 in relation to the total area (Figure 4d). Representative images for each experimental condition are shown in Figure 4e. Fluorescent signals for telo-Collagen1α1 were barely observed in control and starved HCF-V but were significantly elevated after HCF-V cells were stimulated with a pro-fibrotic cocktail containing TGFβ-1, L-ascorbic acid 2-phosphate and dextran sulphate, measured as MFI and fluorescent area quantification (*p* < 0.001 for control and starved conditions compared with stimulated HCF-V). The telo-Collagen1α1 MFI and extracellular secretion signals were slightly lower in cells containing ctrl-SEVs, with both values being significantly lower than in stimulated cells (*p* < 0.01), whereas CD81-SEVs significantly reduced only telo-Collagen1α1 extracellular secretion compared to stimulated cells (*p* < 0.01). Importantly, the treatment of HCF-V cells with both matOSM-CD81-SEVs and mutOSM-CD81-SEVs reduced the telo-Collagen1α1 signal and extracellular deposition significantly compared with stimulated cells only (*p* < 0.01 and *p* < 0.001, respectively). No major differences in MFI were observed when HCF-V cells were treated with matOSM-CD81-SEVs or mutOSM-CD81-SEVs when compared with CD81-SEVs; however, significant differences were observed in terms of extracellular telo-Collagen1α1 deposition (*p* < 0.01 and *p* < 0.001, respectively), suggesting an active role for OSM-enriched SEVs in reducing extracellular telo-Collagen1α1 deposition after fibrosis stimulation in vitro.

These results indicate that native MSC-T-SEVs have a certain intrinsic capacity to counteract fibroblast activation upon stimulation, and adding OSM to these SEVs can boost this effect, making OSM a good candidate to temper cardiac fibrosis after injury.

### 2.5. MutOSM-CD81-SEVs MSC-T Reduce Cardiac Fibrosis in a Model of Isoproterenol-Induced Myocardial Infarction Type 2

Functional comparison of matOSM-CD81-SEVs and mutOSM-CD81-SEVs in vitro revealed no significant differences, although mutOSM-CD81-SEVs were more effective at controlling fibroblast proliferation (Figure 4a). We thus only evaluated mutOSM-CD81-SEVs’ functionality in vivo. We induced cardiac hypertrophy and interstitial fibrosis in mice, using the β-adrenergic agonist isoproterenol (ISO), which has been extensively used in murine models [42]. The pathophysiological effect of ISO mimics myocardial infarction type 2 [43], which occurs when there is a mismatch between oxygen supply and demand. Mice were randomly distributed into the following three experimental groups: ISO + CD81-SEVs, ISO + mutOSM-SEVs, and control (ISO), in which equivalent volumes of PBS were injected (Figure 5a). Mice were injected with 150 mg/kg/day ISO during 5 consecutive days, and SEVs or PBS treatments were administered on days 1 and 7. Animals were euthanized on day 21 for histology.

Heart sections were stained with Picrosirius red, and the collagen area was quantified (Figure 5b,c). No significant differences in the collagen area were observed between mice treated with CD81-SEVs and control mice (ISO-only). By contrast, the collagen-stained area was significantly smaller in mice treated with mutOSM-SEVs than in equivalent control mice and CD81-SEV-treated mice (*p* < 0.05 in both cases). We next evaluated the presence of myofibroblasts 21 days after initiating ISO treatment by staining for the ECM protein periostin. As expected, periostin staining was significantly smaller in mice treated with mutOSM-CD81-SEVs than in mice treated with either CD81-SEVs or ISO only (*p* < 0.05, Figure 5d,e).

We also looked for cardiac hypertrophy, using wheat germ agglutinin staining (WGA) to measure the average area of cardiomyocytes (Figure 6a,b). The results showed that the cardiomyocyte area was significantly smaller in mice treated with CD81-SEVs than in control mice (ISO-only) (*p* < 0.05), and this was more pronounced in mice treated with mutOSM-CD81-SEVs compared to the ISO-only group (*p* < 0.01) and mice treated with CD81-SEVs (*p* < 0.05). In addition, we investigated whether the various EVs treatments could facilitate the reprogramming of macrophages (Mφ) toward a pro-resolutive M2-like phenotype. Notably, the immunofluorescence analysis of the phenotype of infiltrated macrophages in cardiac tissue revealed a lower presence of pro-inflammatory M1 macrophages (F4/F80+-PD-L1+) and a similar abundance of immunomodulatory M2-like macrophages (F4/F80+-CD206+) in animals treated with mutOSM-CD81-SEVs compared to those treated with ISO alone (*p* < 0.05) (Figure 6c,d). Finally, we quantified angiogenesis by using an antibody to CD31. The staining results showed a potent angiogenic effect in mice treated with mutOSM-CD81-SEVs in comparison to mice treated with CD81-SEVs or control mice (Figure 6e,f). We also observed a superior therapeutic effect of mutOSM-CD81-SEVs in terms of vessel density in mutOSM-CD81-SEV-treated mice compared to the control group (*p* < 0.01) and the CD81-SEV-treated group (*p* < 0.001).

Altogether, the results point to an enhanced functional effect of modified OSM on the surface of MSC-T-SEV over native MSC-T-SEVs, as shown by a decrease in collagen deposition and cardiomyocyte hypertrophy, the polarization of Mφ to M2-like phenotype, and an increase in angiogenesis.

## 3. Discussion

OSM (Oncostatin M) is a cytokine with pleiotropic effects that can lead to cardiomyocyte (CM) dedifferentiation, as previously reported [31]. This mechanism provides a competitive advantage in the context of myocardial infarction, as the disassembly of sarcomeres during the dedifferentiation process renders the CM resistant to hypoxia. In fact, studies in animals have shown that treatment with OSM can improve heart function [30]. Additionally, the interplay between OSM and IL-6 in CM can induce further cardioprotection and reinforce the role of OSM in active cardiac remodeling [25]. However, it is important to note that, during therapy, the temporal activation of OSM signaling and the induction of the pathway at physiological doses are critical to achieve a positive outcome. This is because persistent dedifferentiation of cardiomyocytes could contribute to adverse cardiac remodeling, emphasizing the importance of a carefully calibrated approach to OSM-based treatments [30]. For this reason, we thought about delivery of SEVs as a strategy to transiently trigger OSM signaling in the heart. SEVs are nanosized vesicles that are released by many types of cells, including those commonly used for regenerative purposes, such as MSCs and cardiac progenitor cells. Previous work by our group and others demonstrated the feasibility of immortalizing MSCs from different tissues to obtain a stable parental source of MSC-SEVs [44,45]. In particular, the overexpression of human TERT in MSCs and other progenitors does not alter their therapeutic features or the properties of SEVs [46,47]. These cells can be further modified to bolster their therapeutic potential, as performed with wild-type MSCs [48,49,50]. In the present study, we sought to test the effect of OSM expression on the surface of MSC-SEVs by lentiviral transduction of mutated OSM variants resistant to post-translational protease cleavage.

Tetraspanins such as CD63, CD9, TSPAN14, and CD81, along with lamp2b and the C1C2 domain of lactadherin, are commonly utilized proteins to load cargo into small extracellular vesicles (SEVs). Our previous research demonstrated that TSPAN14, CD63, and CD81 fused to the PDGFRB transmembrane (TM) domain were the most efficient proteins for cargo loading [38]. To ensure the successful expression of the recombinant protein on the SEVs surface, we tested the cargo-loading efficiency of OSM mutants fused to the TM domain and three different proteins, namely TSPAN14, the C1C2 domains of lactadherin, and CD81. Although the Western blotting analysis did not reveal any significant differences in cargo efficiency, we ultimately decided to proceed with the fusion to CD81TM, as this protein is commonly expressed in the SEVs of various cell types.

Regarding the cardiovascular system, OSM is related to proliferation, chemotaxis, and tube formation in cultures of human microvascular endothelial cells [26,27,28], but it is also related to the inhibition of cardiac fibroblast proliferation after injury-related stimulation [19,41]. To control OSM expression on MSC-T-SEVs and to track plasmid stability over cell passages, lentiviral vectors included a TetON inducible promoter controlling the expression of CD81, matOSM-CD81, or mutOSM-CD81. The phenotypic characterization of SEVs, using a number of techniques, showed that the matOSM-CD81 and mutOSM-CD81 samples contained a small proportion of SEVs carrying free OSM. The capture of SEVs with CD63, CD9, or CD81 spots revealed similar patterns of SEVs between matOSM-CD81 and mutOSM-CD81 samples, indicative of similar lentiviral transduction efficiencies and SEVs packaging, although the amount of OSM loaded per vesicle was higher in matOSM-CD81, as detected by Western blot. Of note, ~50% of CD63-positive SEVs expressing OSM co-expressed both CD81 and CD9, and only a small proportion expressed only one of the latter, indicating that, upon lentiviral transduction, OSM is sorted in a particular type of SEV. Further studies are warranted to elucidate the relevance of this observation.

We tested the functionality of OSM-expressing SEVs after first evaluating the presence of OSM receptors on the surface of fibroblasts cultured under starving conditions to induce cellular stress. The expression of GP130 and IL31RA, but not IL-31RB, increased as soon as 30 min after starvation, suggesting that matOSM-CD81 and mutOSM-CD81-SEVs can interact with starved cardiac fibroblasts through OSM receptors. Indeed, the ability of SEVs to interact with fibroblast receptors was previously described in the context of exosomes derived from cancer cell lines that expressed TGF-β. In this study, the capability of nanovesicles to induce fibroblast activation and myofibroblast differentiation in vitro was shown [51,52]. The expression of GP130 has also been reported to be triggered in the myocardial tissue of rats subjected to acute myocardial infarction as soon as one day after coronary artery ligation and was maintained for at least two months post-injury [53], which also suggests that SEVs containing OSM on their surface could interact with the afore mentioned receptors in vivo.

We tested OSM-loaded SEVs in vitro to evaluate their capacity to dampen fibroblast proliferation and ECM deposition, finding that both matOSM-CD81-SEVs and mutOSM-CD81-SEVs, but not control SEVs, inhibited cell proliferation and secretion of telo-Collagen1α1 in activated HCF-V cells, pointing to an anti-fibrotic effect of OSM-loaded SEVs. In this context, our analysis did not reveal significant differences in the in vitro effects of matOSM-CD81 and mutOSM-CD81 SEVs when comparing the protein carrying the pro-peptide (PP) domain to that lacking it. However, we did observe a greater decrease in proliferation in starved cardiac fibroblasts when treated with mutOSM-CD81-SEVs compared to matOSM-CD81-SEVs, which supported the use of these type of SEVs over matOSM-CD81-SEVs for the in vivo study.

The isoproterenol infusion model is a commonly employed method for inducing interstitial cardiac fibrosis in mice. [54,55,56]. Fibrosis was quantified in heart sections, using Picrosirius red and periostin staining. Periostin is produced by myofibroblasts after activation of resident fibroblasts [57,58,59,60] in response to stress and during different cardiac pathological conditions, such as myocardial infarction or dilated cardiomyopathy, and is secreted into the extracellular milieu and localizes to areas of fibrosis and collagen accumulation. Animals treated with mutOSM-CD81-SEVs exhibited an increase in endothelial vessel density as compared with control animals. This is in accordance with reports showing the potent angiogenic effect of OSM in the context of inflammation [26,61]. Additionally, the infusion of mutOSM-CD81-SEVs reduced fibrosis, which is consistent with reports showing that OSM plays a role in controlling fibrosis and ECM degradation in the context of cardiac repair after injury [19,34]. In a different setting, the presence of OSM during the healing of keloid or hypertrophic scars modulated the secretion of ECM components and excessive scarring [41]. Of note, although treatment with CD81-SEVs and Ctrl-SEVs reduced the accumulation of telo-Collagen1α1 in vitro, we did not observe a significant anti-fibrotic effect in vivo in regard to angiogenesis or cardiomyocyte hypertrophy. However, other studies have reported the anti-fibrotic and pro-angiogenic properties of wild-type MSC-SEVs in the context of myocardial ischemia [5,62,63,64,65]. The lower doses used in the present study may account for these discrepancies and underline the effectiveness of the proposed strategy to boost the therapeutic potential of MSC-SEVs.

The infusion of mutOSM-CD81-SEVs in ISO-treated animals also exerted a positive effect in cardiomyocytes, as the analysis of cardiac muscle fibers in control and mutOSM-CD81-SEV-treated animals revealed the capacity of these vesicles to prevent cardiac hypertrophy. This is in accordance with the previously mentioned works demonstrating the ability of OSM to maintain the dedifferentiated cardiomyocyte state and to prevent the distortion of cardiac architecture and the increase in cardiomyocyte length and width through limited activation of OSM receptors in cardiac muscle cells [25,30]. Although the activation of OSM signaling cascades is protective under acute stress conditions, its chronic activation contributes to the development of heart failure [25]. The in vivo retention of SEVs in tissues after infusion is low unless they are encapsulated in polymeric matrices [66,67], and they are rapidly cleared by the innate immune system [68]. This process prevents the risk of chronic activation of OSM cascades by mutOSM-CD81-SEV infusion. Our findings provide evidence that even a brief period of retention of small extracellular vesicles (SEVs) in the circulation is enough to trigger a shift in the polarization of macrophages (Mφ) toward an M2-like phenotype and facilitate the pro-resolutive phase. Furthermore, research has shown that both peritoneal and resident cardiac macrophages in mice exhibit comparable transcriptional profiles, including the presence of OSM receptors that are known to play a role in inducing angiogenesis [69,70]. Thus, the i.p. injection of mutOSM-CD81-SEVs could explain the cardioprotective effect exerted by the treatment. Despite our significant findings, there are some limitations to this study that should be noted. Firstly, we did not conduct biodistribution studies, which limits our ability to draw conclusions regarding the localization and distribution of SEVs in vivo. Additionally, our histological analysis of M1/M2 macrophages in cardiac tissue did not allow us to differentiate between resident cardiac macrophages and those that infiltrate from circulating monocytes. As a result, we cannot conclusively determine whether the OSM-related signaling pathway is activated through direct interaction of OSM-expressing SEVs with cells in heart tissue, such as cardiomyocytes or endothelial cells, or via activation of peritoneal macrophages that may serve as the primary mediators of the reparative process. In conclusion, we generated a genetically modified MSC cell line able to release SEVs expressing functional OSM proteins on their surface, which can promote cardiac restorative processes. These findings might guide the development of refined MSC-SEV-related strategies for heart repair.

## 4. Materials and Methods

### 4.1. Prokaryote Cells

Douglas Hanahan 5α™ (DH5α) competent cells were obtained from Thermo Fisher Scientific (Waltham, MA, USA). The genotype is F-*Φ80lac*ZΔM15 Δ(*lac*ZYA-*arg*F) U169 *rec*A1 *end*A1 *hsd*R17(rk-, mk+) *pho*A *sup*E44 *thi*-1 *gyr*A96 *rel*A1 λ-. Colony selection after bacterial transformation was performed using ampicillin (Thermo Fisher Scientific).

### 4.2. Eukaryotic Cell Lines

Human embryonic kidney (HEK 293T) cells were obtained from the American Type Culture Collection (Manassas, VA, USA). Cells were cultured in high-glucose DMEM containing 10% heat-inactivated fetal bovine serum (FBS) and 1% penicillin/streptomycin solution (P/S), all from Gibco (ThermoFisher Scientific). Expi293F Expression System (Gibco, ThermoFisher Scientific) was used for transient expression studies in mammalian 293 cells, following the manufacturer´s indications. Dental pulp MSCs in passage 1/2 were obtained from the Spanish National Cell Line Bank (BNLC) through the Inbiobank Foundation (San Sebastián, Spain). MSC were immortalized and characterized in our laboratory through lentiviral transduction with telomerase reverse transcriptase (MSC-T) [46]. MSC and MSC-T were cultured in low-glucose DMEM containing 10% heat-inactivated FBS and 1% P/S, all from Gibco. HCF-V cells were obtained from PromoCell and cultured in Fibroblast Growth Medium-3 BulletKit™ (PromoCell, Heidelberg, Germany). All cells were cultured in a Forma Series II Incubator Model 3141 (ThermoFisher Scientific).

### 4.3. Plasmid Design

Fusion proteins were designed to anchor two versions of OSM to CD81 tetraspanin. Mutant OSM (mutOSM) consisted of the full sequence of human OSM but including a specific mutation on the proteolytic CS between the main chain and the PP domain of the sequence, resulting in a substitution of an arginine (R) with a glycine (G). The matOSM construct contained the mutation, but not the PP domain, whereas the mutOSM construct contained the full sequence of OSM, maintaining the previous specified mutation. Both variants were introduced in the plasmids next to a transmembrane domain (PDGFRB, TM) located at the N-terminus of CD81to maintain modified OSM on the outer part of SEVs’ membrane. A third plasmid containing only CD81TM was included as control. The expression of CD81, matOSM-CD81, and mutOSM-CD81 was controlled by a Tet-On inducible promoter, which responds to doxycycline. Green fluorescent protein (GFP) was included in the three vectors, and its expression was controlled by a constitutive promoter (EF1α). Plasmids were obtained from SBI Gene Universal (Gene Universal Inc., Newark, DE, USA).

### 4.4. Bacteria Transformation and Amplification

Plasmids were transformed and amplified in DH5α™ bacteria, using standard heat-shock protocols. Briefly, bacteria were mixed with the correspondent DNA plasmids, and bacteria were transformed in a pre-warmed water bath (42 °C) for 45 seconds and placed on ice for 2 minutes. Then 1 mL of pre-warmed lysogeny broth (LB; Sigma-Aldrich, St. Louis, MO, USA) medium was added, and samples were incubated at 180 rpm in an orbital shaker at 37 °C for 1 h. Samples were then centrifuged at 400× *g* for 1 min, and the pellet was resuspended in 100 μL of LB. The resulting mixture was added to LB-agar Petri dishes with ampicillin for bacterial clone selection and incubated in a bacteriological incubator overnight.

### 4.5. Isolation and Purification of Plasmid DNA from Bacteria

Colonies were individually picked in tubes containing LB with ampicillin and incubated in an orbital shaker at 180 rpm and 37 °C for 6 to 8 h. The Plasmid MiniPrep Kit (GE Healthcare, Chicago, IL, USA) was used for DNA isolation from bacteria. For larger amounts of DNA, transformed bacteria were further grown in flasks with 200 mL LB containing ampicillin at 180 rpm and 37 °C overnight, and the Jetstar Plasmid Maxi Prep Kit (Gentaur, Malaga, Spain) was used for DNA isolation. DNA was quantified by spectrophotometry (NanoDrop 1000, ThermoFisher Scientific).

### 4.6. Engineered MSC-T Generation

Genetically modified MSC-T cells for CD81, matOSM-CD81, and mutOSM-CD81 overexpression contained a Tet-On inducible vector system whose expression depends on the presence of doxycycline (10 µg/mL; Sigma-Aldrich). Lentiviral-transduced MSC-T cells were seeded under the same conditions, but standard FBS was replaced for heat-inactivated tetracycline-free FBS (BioWest, Nuaillé, France) to allow the expression of the vector encoding proteins only when doxycycline was added to the medium. HEK 293T cells were used as packaging cells to produce lentiviral particles, using third-generation lentiviral transduction protocol [71]. After titration, an adequate multiplicity of infection of viral particles was used to transfect MSC-T, and 8 µg/mL polybrene (Sigma-Aldrich) was added to increase infection rate.

### 4.7. Isolation of Small Extracellular Vesicles

The SEVs RM contained low-glucose DMEM with 10% tetracycline and vesicles-free FBS and 1% P/S. Doxycycline (Sigma-Aldrich) was added at a final concentration of 10 µg/mL and refreshed every day. Vesicles naturally present in FBS were removed by ultracentrifugation at 150,000× *g* for 16 h at 4 °C, followed by filtration through a polyether sulfone membrane filter with a pore size of 0.22 µm, as described [72]. Cells were incubated in SEV RM with doxycycline for 48 h before supernatant collection for SEV isolation. Isolations were performed in batches of 250 mL of supernatant.

SEVs’ isolation from RM was performed by differential ultracentrifugation and filtration. All steps of this protocol were performed at 4 °C. First, the supernatant was centrifuged at 2500× *g* for 20 min to precipitate and remove remaining cells. Supernatants were transferred to ultracentrifuge tubes and centrifuged at 15,000× *g* for 45 min to pellet the cellular debris. The supernatant was again collected and filtered (pore size 0.2 µm) to exclude large EVs. Thereafter, the filtered supernatant was transferred to clean tubes and ultracentrifuged at 100,000× *g* for 2 h. Subsequently, the supernatant was completely removed, and each pellet was resuspended in 1 mL of cold PBS. The resulting solutions from the same biological conditions were pooled in only one tube for a final washing step, which was achieved by a second ultracentrifugation at 100,000× *g* for 2 h. Finally, the supernatant was removed, and the SEV pellet was resuspended in the same volume of the desired solution.

The protein content of isolated SEV samples was measured using the BCA assay (Pierce™ BCA Protein Assay Kit, Thermo Fisher Scientific). The amount of SEV protein used for in vitro experiments was 30 µg/mL.

### 4.8. Transmission Electron Microscopy

SEV pellets collected from equal amounts of culture medium were suspended in 200 µL of PBS, loaded onto pure carbon-coated copper grids, fixed with 2% PFA and 1% glutaraldehyde, contrasted with 1% uranyl acetate, and embedded in 0.4% methyl cellulose for analysis by negative staining. The grids were examined with a FEI Tecnai G2 Spirit transmission electron microscope (Thermo Fisher Scientific). All images were acquired using a Morada digital camera (Olympus Soft Image Solutions GmbH, Münster, Germany).

### 4.9. Nanoparticle Tracking Analysis

NTA was used to determine the SEV concentration and particle size distribution. All particle tracking analyses were performed using a LM14C Nano Sight instrument (Malvern Instruments Ltd., Malvern, UK), using the same settings (camera level 16, 3 videos of 90 s and 1300/512 slider shutter/gain, respectively), at the proper concentration, to obtain around 50 particles/frame. An analysis of the acquired videos was performed with threshold 5 and gain 12 with the proprietary NTA v3.2 software.

### 4.10. Single-Particle Fluorescence and Interferometry Imaging with ExoView^®^

ExoView*^®^* technology is based on single-particle interferometric reflectance imaging detection of SEVs captured on a silicon substrate chip, constituted by an array of spots printed with different antibodies. This can be combined with EV immunostaining with fluorophore-conjugated antibodies, followed by fluorescence imaging, and can be multiplexed for the simultaneous detection of up to three distinct fluorophores [38]. Isolated SEVs were diluted 1:10 in incubation solution (1×); 50 μL of this sample was carefully pipetted onto the silicon chip coated with individual antibody spots against human CD9, CD63, and CD81, as well as negative isotype controls. After overnight incubation in a 24-well plate, chips were washed three times on an orbital shaker orbital at 500 rpm for 3 min with Solution A. Then the chips were incubated as before at 500 rpm for 1 h at room temperature with a cocktail of fluorescent antibodies (anti-CD81-AF488, anti-CD9-AF555, and anti-OSM-AF647) diluted in Blocking Solution. Chips were washed once in Solution A, three times in Solution B, and once in deionized water. Chips were carefully removed from the 24-well plate, washed further in deionized water, and removed for drying. Image and data acquisition for each chip was performed with the ExoView^®^ R100 (NanoView Biosciences), and a data analysis was performed with ExoView Analyzer 3.1.4. An antibody to OSM was purchased from Thermo Fisher (Cat# #MA5-23810, clone: 17001). Fluorescent conjugation was performed with Alexa Fluor^®^ 647 Conjugation Kit (ab269823; Abcam, Cambridge, MA, USA).

### 4.11. HCF-V Phenotype Stimulation In Vitro

HCF-V were seeded at 25,000 cells/cm^2^ and were allowed to attach for 24 h. Thereafter, culture medium was replaced with starvation medium (high-glucose DMEM, 0.5% vesicle-depleted FBS, 1% non-essential amino acids, and 1% P/S). After 16 h, fibroblasts were stimulated with 100 μM L-ascorbic acid 2-phosphate (Sigma-Aldrich), 3.16 ng/mL recombinant TGFβ-1 (R&D Systems, Minneapolis, MI, USA), and 3.16 μg/mL dextran sulphate (Sigma-Aldrich), as described [73]. Treatments were added at the same time as the media exchange. Unstimulated cells (cells starved but not stimulated) and untreated cells (neither starved nor stimulated, control) were used as control conditions. HCF-Vs were stimulated for 48 h before proceeding to readouts.

### 4.12. Bromodeoxyuridine Cell Proliferation Assay

Bromodeoxyuridine (BrdU) is a pyrimidine-analogue synthetic nucleotide that can be used to label nascent DNA in viable cells. During the process of DNA replication, BrdU can replace thymidine and incorporate into the synthesized DNA of actively dividing cells. First, cells were seeded in 18 mm coverslips in 24 multi-well plates in low-glucose DMEM supplemented with 10% tetracycline-free FBS, 1% P/S at a density of 50,000 cells/cm^2^. On the next day, BrdU reagent (10 μg/mL, Life Technologies, Carlsbad, CA, USA) was diluted in culture medium containing the corresponding treatments and incubated for 8 h. After this time, cells were washed 3 times with PBS and fixed with ice-cold 70% EtOH at 4 °C for 10 min. Coverslips were washed 3 times with PBS and incubated with 1N HCl at 37 °C for 30 min. HCl was used to unmask BrdU incorporated in newly synthetized DNA. For acid neutralization, borate buffer (nM: 100 boric acid, 75 sodium chloride, 25 sodium tetra-borate in distilled water; pH adjusted to 8.3) was added and samples were incubated at room temperature for 10 min. Anti-BrdU (Abcam, Cambridge, UK) 1:100 was used to detect BrdU incorporation by the cells. Images of random fields were taken using an upright fluorescence microscope (Model DM6000B, Leica, Wetzlar, Germany), and a total of 100 nuclei were counted in each sample. The proliferation rate was calculated as the ratio of double-stained DAPI and BrdU cells (replicative cells) and single-stained DAPI cells (non-replicative cells).

### 4.13. Migration Assay

Cells were seeded into 24-well-plate transfer inserts (pore size: 8 μm; BD, Sunnayvale, CA, USA) in basal culture medium (low-glucose DMEM + 0.5% FBS TTC free + 1% P/S), at a density of 10,000 cells/cm^2^. After cell stabilization, the lower chamber medium was replaced with basal culture medium supplemented with recombinant human OSM (10 ng/mL). Basal medium was used as migration negative control. After 8 h, non-migrated cells were removed from the upper side of the Transwell inserts with cotton swabs, and cells in the lower part of the Transwell insert were fixed with ice-cold 70% EtOH for 10 min at 4 °C. Then membranes were washed with PBS, stained with 1 μg/mL DAPI, cut, and transferred to microscope slides (Menzel Gläser, Afora, Braunschweig, Germany). Images were taken in an upright fluorescence microscope (Model DM6000B, Leica), and the number of nuclei on each sample was counted.

### 4.14. Western Blotting

Protein samples from cell lysates or SEVs were separated by 10% SDS-PAGE and transferred to PVDF membranes (Thermo Fisher Scientific). Thereafter, PVDF membranes were blocked with 5% dry milk in Tris-buffered saline (20 mM Tris pH 7.5, 150 mM NaCl). Primary antibodies diluted in blocking solution were incubated at 4 °C overnight. The following primary antibodies were used: α-hOSM, (MAB295, R&D Systems), α-GP130 (#3732, Cell Signaling Technology, Danvers, MA, USA) α-IL-31RA (ab113498, Abcam), and α-IL-31-RB (10982-1-AP, Proteintech, Rosemont, IL, USA). Antibodies were used at a 1:200 dilution. Non-bound primary antibody was removed by three washing steps with PBS. Blots were then incubated with IgG-HRP secondary antibodies for 1 h at room temperature (Promega), and signals were detected using the Amersham ECL Western Blotting Detection Kit (GE Healthcare). Tubulin α-1A (AA13, Sigma-Aldrich) signal was used to standardize cellular protein quantities. Densitometric analysis was performed with ImageJ software (NIH).

### 4.15. Immunofluorescence

HCF-V cells were fixed with 4% PFA for 10 min at 37 °C. After washing with PBS, cells were permeabilized with 0.2% Triton in PBS for 1 h at room temperature and blocked for nonspecific epitopes with 2% BSA in PBS (both from Sigma-Aldrich). Thereafter, samples were incubated with primary antibodies solution, containing 0.1% BSA in PBS. The following antibodies were used: α-Ki-67 and α-teloCollagen1α1 (ab16667 and ab241825, both from Abcam; 1:100 dilution). Next, samples were washed 3 times with PBS and incubated with an α-rabbit Alexa Fluor 488 secondary antibody (Life Technologies) and DAPI for 1 hour at room temperature. Images were taken using CellVoyager CV8000 high-throughput screening system (Yokogawa, Tokyo, Japan), and fluorescent signals were analyzed with Leica Application Suite Version 2.4.0 R1 version and ImageJ 1.53t software (NIH).

### 4.16. Isoproterenol In Vivo Model

All experimental procedures involving the use of animals were approved by the institutional ethical and animal care committees according to guidelines from Directive 2010/63/EU of the European Parliament on the protection of animals used for scientific purposes, enforced in Spanish law under Real Decreto 1201/2005. Regional government (Generalitat Valenciana) authorized the procedure 2021/VSC/PEA/0214 for the model of isoproterenol-induced heart failure. Experimental procedure is illustrated in Figure 5a. Adult C3H/HeNCrl male mice (10–14 weeks old, weighting 35–40 g) were selected for in vivo studies (Charles River Laboratories, Wilmington, MA, USA). Mice were randomly distributed into three experimental groups (*n* = 5 in each group): control (ISO), mice treated with CD81-SEVs (ISO + CD81-SEVs), and mice treated with mutOSM-CD81-SEVs (ISO + mutOSM-CD81 SEVs). Isoproterenol (I5627, Sigma Aldrich, St. Louis, MO, USA) was subcutaneously infused once a day for 5 consecutive days, using a dose of 150 mg/kg/day). Intraperitoneal injection of 100 µg of SEV protein was performed just after the first isoproterenol injection. One week later, mice received a second intraperitoneal SEV injection (50 µg of SEV protein). Equivalent volumes of PBS were injected in the ISO group. Euthanasia was conducted after 21 days, and cardiac tissues were resected for further analysis.

### 4.17. Histology

Mice hearts were collected at day 21 after isoproterenol treatment, fixed in 2% PFA, embedded with paraffin, and sectioned into 5 µm slices for histological analysis by Picrosirius red and WGA staining. One part of the tissue sections was stained for 30 min in a solution of 0.1% Sirius red in saturated aqueous picric acid for collagen bundle staining, as described [74]. Samples were prepared and images captured with a Leica DMD108 microscope. ImageJ was used for whole-collagen-area quantification. A second batch of tissue sections were stained with 1 µg/mL WGA conjugated to Texas Red-X (Invitrogen, Thermo Fisher Scientific, Waltham, MA, USA). To visualize cell nuclei, DAPI staining was used, and the slides were mounted with FluorSave™ Reagent (Merck Millipore, Burlington, MA, USA). Images were captured with a Leica DM2500 fluorescence microscope and analyzed by ImageJ for the cross-sectional area of cardiomyocytes. For the immunofluorescence analysis of M1 and M2-like macrophages, slides were initially blocked with 10% fetal bovine serum (FBS) in phosphate-buffered saline (PBS) for 1 h. Heat-induced antigen retrieval was performed before antibody labeling. The slides were incubated with primary antibodies, including mice anti-F4/F80 (dilution 1/200, Abcam, ab6640), rabbit anti-CD206 (dilution 1/200, Abcam, ab64693), or rabbit anti-CD274 (dilution 1/200, AB Clonal A11273), overnight, in a humidified chamber, at 4 °C. Subsequently, the slides were washed three times for 5 minutes each in PBS. The secondary antibodies, including anti-rat IgG Alexa 555, or anti-rabbit IgG Alexa 488, were applied for 1 hour, followed by three washes for 5 minutes each in PBS. To visualize cell nuclei, DAPI staining was used, and the slides were mounted with FluorSave™ Reagent (Merck Millipore). The sections were observed using a fluorescent microscope, and the final image processing and quantification were performed using ImageJ software by counting green and red spots in a defined area. Vessel density of cardiac tissue was assessed by staining tissue sections with an α-CD31 antibody (dilution, 1/200; sc-1506; clone ID, M-20; Santa Cruz Biotechnology, Dallas, TX, USA). Heat-induced antigen retrieval (HIER) was conducted prior to antibody labeling. Images were captured with a Leica DM2500 fluorescence microscope and analyzed in ImageJ to measure the number of vessels/mm^2^. For the assessment of perivascular fibrosis, tissue sections were stained with anti-POSTN (dilution 1/100, 19899-1-AP, Proteintech) and anti-αSMA (dilution, 1/100; clone 1A4; M0851; Dako). HIER and permeabilization of the tissue with Tween-20 were performed before antibody incubation. To visualize cell nuclei, DAPI staining was used, and the slides were mounted with FluorSave™ Reagent (Merck Millipore). Images were captured with a Leica DM2500 fluorescence microscope and analyzed in ImageJ to measure the perivascular area covered with POSTN.

### 4.18. Statistical Analysis

Three biological replicates (*n* = 3) were included for all data of in vitro studies. Five animals were included in each experimental group for the in vivo isoproterenol model. Data from all experimental groups were compared using ANOVA, followed by Tukey’s post hoc test. All results are presented as mean ± SD. Asterisks indicate statistically significant different between experimental conditions, where *p* < 0.05 is indicated by *, *p* < 0.01 by **, and *p* < 0.001 by ***. Differences were considered at *p* < 0.05 with a 95% confidence interval. Statistical analysis was performed using GraphPad Prism 8 software (La Jolla, CA, USA).

## Figures and Tables

**Figure 1 ijms-24-06467-f001:**
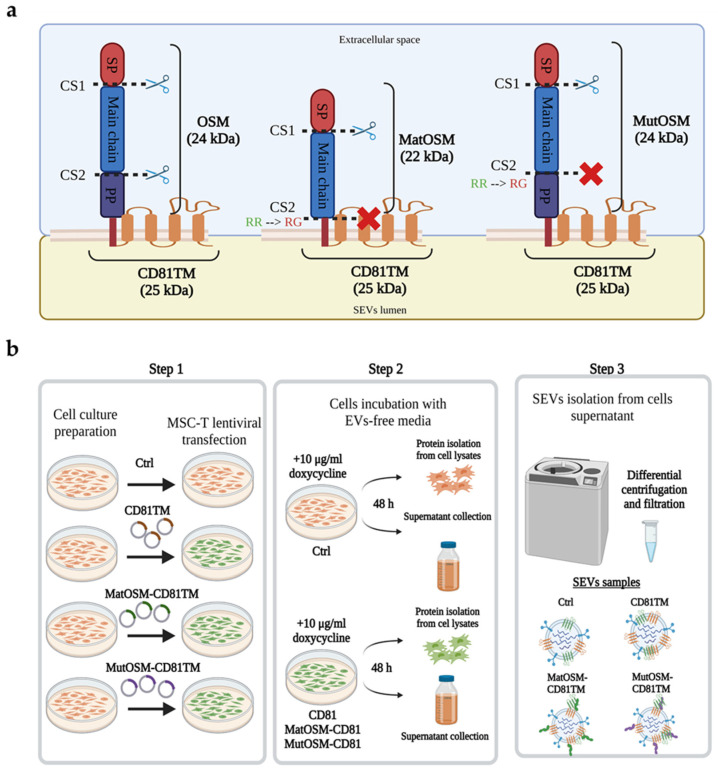
Generation of OSM-loaded MSC-T-SEVs through OSM sequence modification and fusion to CD81TM. (**a**) Structure scheme of native and recombinant proteins with introduced variants in the OSM sequence. The PP domain sequence was removed in mature OSM (matOSM), and a nucleotide mutation was introduced to change an R for a G in CS2 to block excision of matOSM from SEV membranes. The entire OSM sequence was maintained in mutant OSM (mutOSM), but the same R-to-G mutation in CS2 was introduced. Both matOSM and mutOSM sequences are anchored to CD81TM to favor their presence on SEV membranes. (**b**) General experimental procedure to obtain engineered MSC-T and SEV samples. Step 1: Cell culture preparation for lentiviral transfection. Lentiviruses containing CD81, matOSM-CD81, or mutOSM-CD81 were used to generate engineered MSC-T. Step 2: MSC-T expansion and incubation with EVs-free medium containing doxycycline for 48 h for supernatant collection and protein isolation from cell lysates. Step 3: SEV isolation from cell supernatants by ultracentrifugation and filtration. Fusion of two proteins is represented with a dash (-). Scissors represent where OSM cleavage occurs, while red crosses represent where the excision does not occur. OSM, Oncostatin M; SP, signal peptide; CS, cleavage site; PP, propeptide; R, arginine; G, glycine: MSC-T, immortalized mesenchymal stromal cells; EVs, extracellular vesicles; SEVs, small extracellular vesicles. Image was created using BioRender.

**Figure 2 ijms-24-06467-f002:**
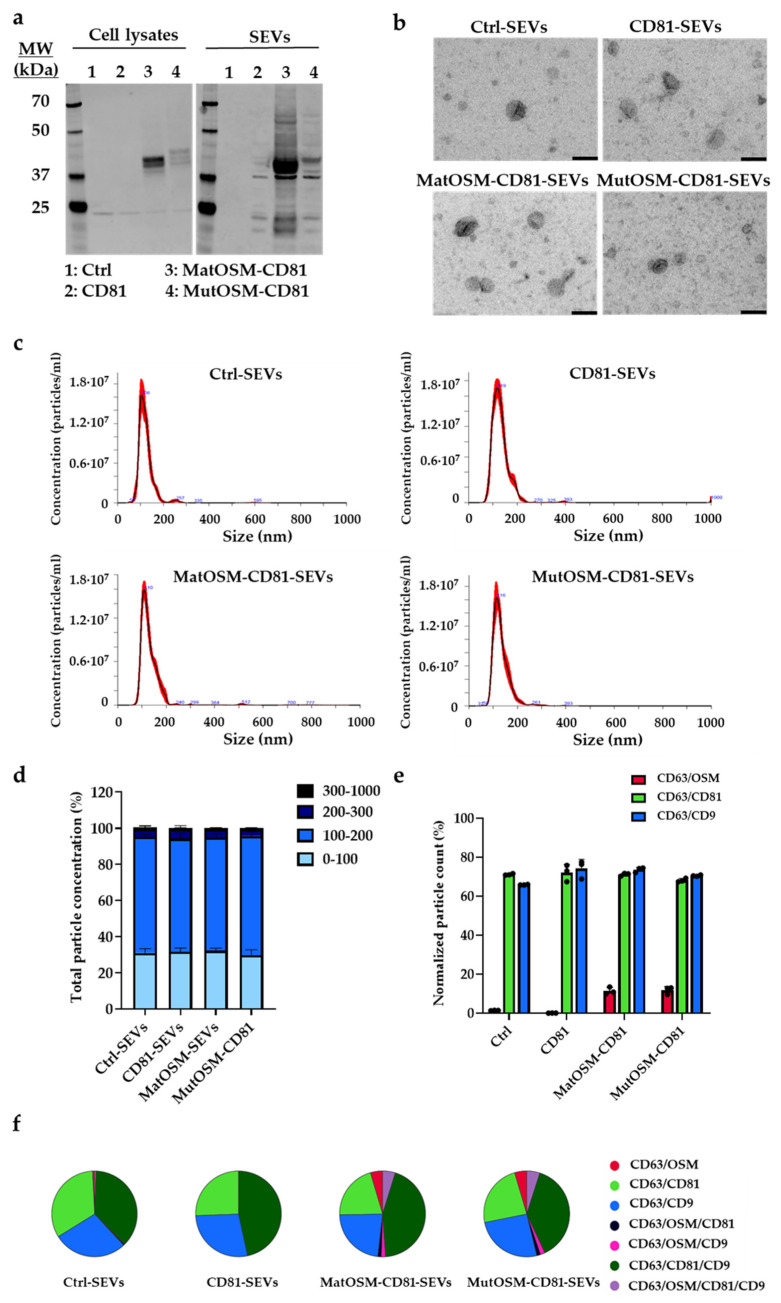
SEVs derived from genetically modified MSC-T cells carry the recombinant proteins mat/mutOSM-CD81 on their surface but maintain the same physical characteristics in terms of morphology and size distribution. (**a**) OSM detection in protein samples of cell lysates or SEVs from MSC-T (Ctrl) (lane 1), MSC-T-CD81 (CD81) (lane 2), MSC-T-matureOSM-CD81 (matOSM-CD81) (lane 3), and MSC-T-mutantOSM-CD81 (mutOSM-CD81) (lane 4). OSM was identified at both native molecular weight (22–24 kDa) and its expected molecular weight after fusion to CD81 (45–47 kDa). SEV protein extracts, 10 µg/lane; cell lysate protein extracts, 30 µg/lane. (**b**) Representative images of SEVs obtained by transmission electron microscopy (scale bar = 100 nm). (**c**,**d**) Nanoparticle tracking analysis, showing average particles size (nm) and particles size distribution histograms. (**e**) ExoView analysis showing percentage of OSM-, CD81-, or CD9-positive SEVs in samples after capture with a CD63 antibody. (**f**) Both the proportion of SEVs double-positive for CD63 and other proteins (OSM, CD81 and CD9) and triple-positive for these markers was analyzed in all SEVs experimental groups and plotted as pie charts. Each experiment was performed three times, and data were analyzed by ANOVA and Tukey’s post hoc test, using values from control cells or SEVs as the reference and represented as mean ± SEM (ns: non-significant).

**Figure 3 ijms-24-06467-f003:**
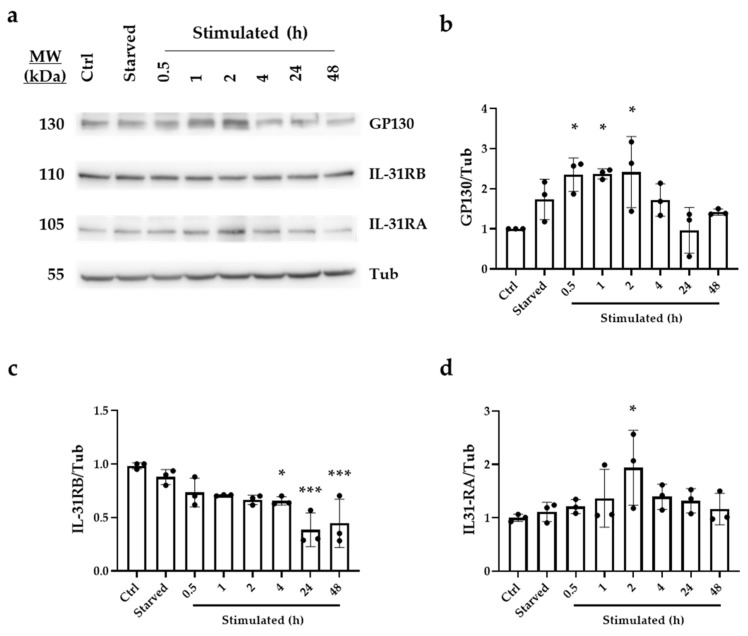
Protein levels of the OSM receptors LIF-R/GP130 and IL-31RA are increased after HCF-V stimulation in vitro. (**a**) Representative Western blots for GP130, IL-31RA, and IL-31RB and β-tubulin (Tub) expression in HCF-V cells were measured under basal conditions (Ctrl), after starvation (Starved) and after starvation and stimulation with a pro-fibrotic cocktail containing L-ascorbic acid 2-phosphate, dextran sulphate, and recombinant TGFβ-1. Samples were collected both after starvation and after 0.5, 1, 2, 4, 24, and 48 h of starvation and fibrosis stimulation. Untreated cells in basal conditions were used as control. (**b**) GP130 protein expression quantification. (**c**) IL-31RB protein expression quantification. (**d**) IL-31RA protein expression quantification. Protein content was measured by densitometry in ImageJ 1.53t, using Tub as a loading control for each condition. Results were obtained from three independent experiments and were analyzed with ANOVA and Tukey’s post hoc test, using values from ctrl cells as a reference and represented as mean ± SEM (* *p* < 0.05; *** *p* < 0.001).

**Figure 4 ijms-24-06467-f004:**
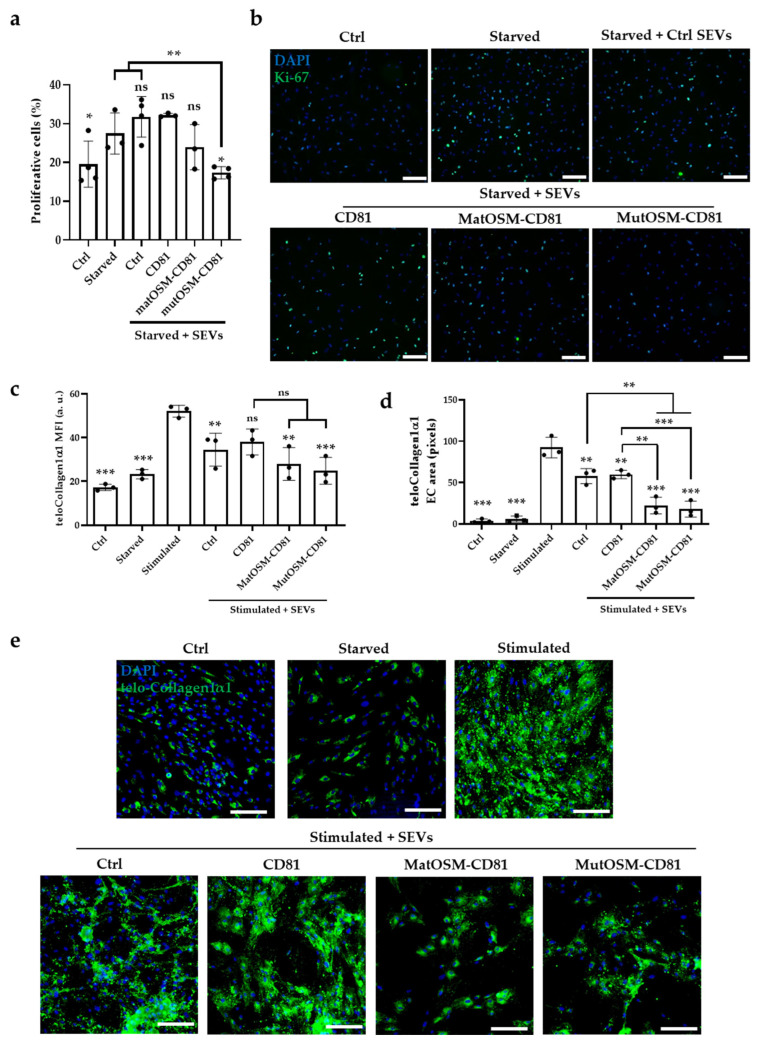
MutOSM-CD81-enriched SEVs have an enhanced ability to reduce HCF-V proliferation and telo-Collagen1α1 expression and secretion to the extracellular space. (**a**) Quantification of HCF-V proliferation rate under basal conditions (Ctrl), after starvation (starved), and after treatment with SEVs. The ratio between number of proliferative cells (Ki-67-positive nuclei, in green) and total number of cells (DAPI-positive nuclei, in blue) is shown. (**b**) Representative images of Ki-67 and DAPI nuclei staining for each experimental condition. (**c**,**d**) Quantification of telo-Collagen1α1 protein, which is shown in green. Mean fluorescence intensity (MFI) normalized to number of cells and extracellular telo-Collagen1α1 area are shown. (**e**) Representative images of telo-Collagen1α1 detection (shown in green) in samples under all experimental conditions. DAPI was used for nuclei staining (shown in blue). Scale bar = 200 µm. Both starved and stimulated cells were treated with equivalent volumes of vehicle (PBS). All results were obtained from three independent experiments and were analyzed with ANOVA and Tukey’s post hoc test. All experimental conditions were compared with both starved cells and starved cells plus ctrl-SEVs for statistical analysis in proliferation experiments. Values for stimulated cells and stimulated cells treated with ctrl-SEVs were used as a reference to compare with the other conditions for the in vitro fibrosis model and are represented as mean ± SEM (ns, non-significant differences; * *p* < 0.05, ** *p* < 0.01, and *** *p* < 0.001).

**Figure 5 ijms-24-06467-f005:**
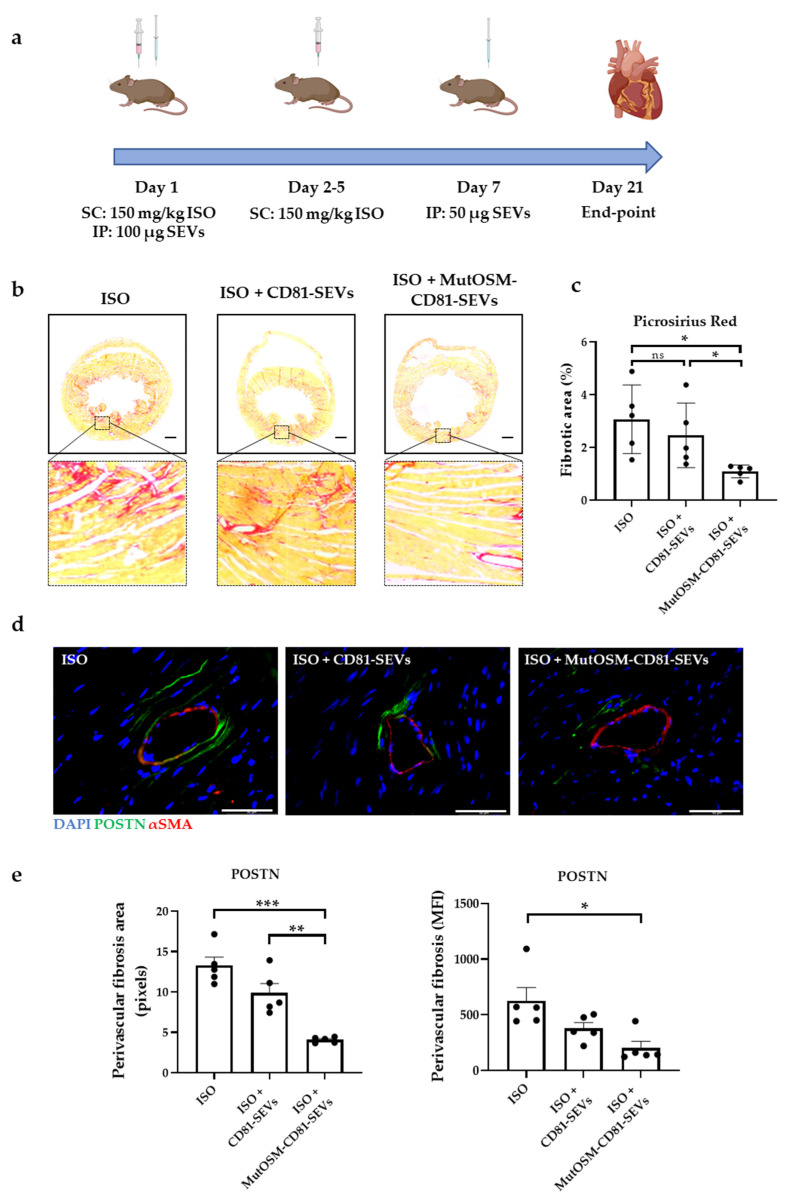
MutOSM-CD81 SEVs reduce collagen deposition in an isoproterenol (ISO) in vivo model of myocardial infarction type 2. (**a**) Schematic of the experimental procedure. Three experimental groups were included: ISO (ctrl), ISO + CD81-SEVs, and ISO + mutOSM-CD81-SEVs. Fibrosis was induced by a daily ISO subcutaneous (SC) injection (dose: 150 mg/kg) during 5 consecutive days. Two intraperitoneal (IP) doses of SEVs (CD81 or mutOSM-CD81) were injected (day 1 and day 7). An equivalent volume of vehicle (PBS) was injected in the ISO-only experimental group. (**b**) Representative images of Picrosirius red staining. Scale bar: 200 µm. (**c**) Quantification of fibrotic area in Picrosirius red-stained heart sections. The percentage of collagen-covered area related to total heart area is represented for each animal. (**d**) Representative images of periostin (POSTN) and α-smooth muscle actin (αSMA) double staining. Scale bar: 50 µm. (**e**) Quantification of perivascular fibrosis by POSTN staining in pixels and mean fluorescence intensity. Five animals were included in each group. ImageJ was used for image analysis and colorimetric and fluorescence quantification. Data were analyzed with ANOVA and Tukey’s post hoc test, using values from ctrl group (ISO) as reference conditions and represented as mean ± SEM. Data from CD81-SEVs and mutOSM-CD81-SEVs were also compared (ns, non-significant; * *p* < 0.05, ** *p* < 0.01, and *** *p* < 0.001).

**Figure 6 ijms-24-06467-f006:**
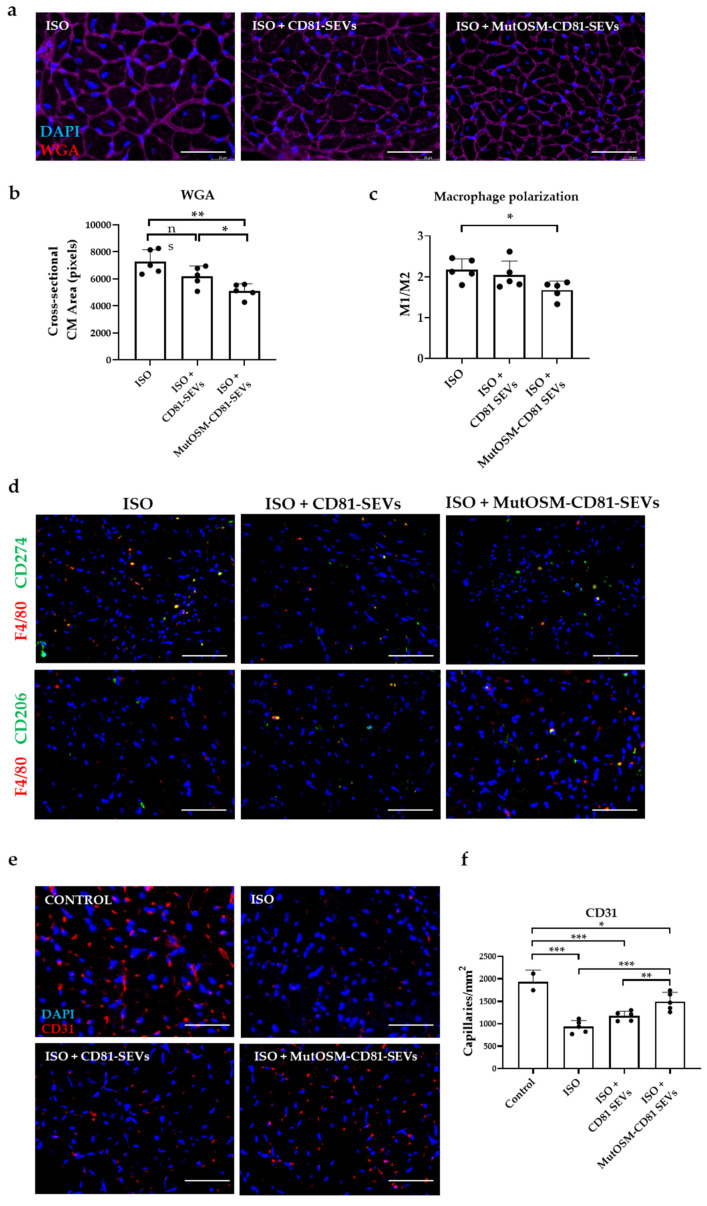
MutOSM-CD81 SEVs reduce cardiac hypertrophy and increase angiogenesis in an isoproterenol (ISO) in vivo model of myocardial infarction type 2. (**a**) Representative images of wheat germ agglutinin (WGA) staining (shown in red) and nuclei staining (DAPI, shown in blue) for each experimental group. Scale bar: 50 µm. (**b**) Measurement of cardiomyocyte area, using WGA staining in heart sections of animals. (**c**) Immunodetection of F4/F80 (pan-macrophage marker; red) and CD274 (PD-L1; pro-inflammatory Mφ1 marker; green) or CD206 (pro-regenerative Mφ2 marker; green) in heart samples 21 days after ISO treatment. Scale bar: 50 μm. (**d**) Quantification of double-positive cells per mm^2^. Five sections of 0.14 mm^2^ per mouse were analyzed. (**e**) Measurement of microvasculature, using an antibody to CD31. Circular stained structures with an area up to 23.5 µm^2^ were counted. (**f**) Representative images of CD31 staining (shown in red) and nuclei staining (DAPI, shown in blue) for each experimental group. Scale bar: 50 µm. Five animals were included in each group. ImageJ was used for image analysis and colorimetric and fluorescence quantification. Data were analyzed with ANOVA and Tukey’s post hoc test, using values from the control group (ISO) as reference conditions and represented as mean ± SEM. Data from CD81-SEVs and mutOSM-CD81-SEVs were also compared (ns, non-significant statistic differences; * *p* < 0.05, ** *p* < 0.01, and *** *p* < 0.001).

## Data Availability

All data generated and/or analyzed during this study are included in this published article and its additional files. All the data can be shared upon request by email.

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
