# Peer review of "Oncostatin M-Enriched Small Extracellular Vesicles Derived from Mesenchymal Stem Cells Prevent Isoproterenol-Induced Fibrosis and Enhance Angiogenesis"

_ijms, 2023, doi:10.3390/ijms24076467_

Round 1

Reviewer 1 Report (Previous Reviewer 1)

I would like to thanks authors for accepting and answering all my comments. I am happy with all changes introduced in this new version. I just want to point out that it looks that blot image regarding IL-31RA in figure 3a is missing.

Author Response

I am so sorry about the mistake, apparently I did not paste the figure properly. Here is a new version with WB of IL-31RA. I also modified the text in page 16 regarding the timepoint of significant changes of IL-31RA expression, since the corrections regarding the previous version were also missed.

Sincerely

Reviewer 2 Report (Previous Reviewer 4)

The authors have responded to concerns and the manuscript has been appropriately improved.

Author Response

Thank you for your decision

Sincerely

This manuscript is a resubmission of an earlier submission. The following is a list of the peer review reports and author responses from that submission.

Round 1

Reviewer 1 Report

First of all, I would like to congratulate authors for such an elegant work to demonstrate the value of genetically modified proteins in extracellular vesicles to have a potential therapeutic roles in cardiovascular diseases. 

I just have one major concern and some minor comments.

-  The major concern comes within figure 3. A minimum number of experiments is required. Please do complete up to at least 3 biological replicates.

- Images from Fig. 5d will benefit by adding a vascular cell marker (ie. SMA or CD31).

- Please do review mentions to supp. figs since it looks that at least supp. fig 2 at page 4 should be 3a, right? 

- Please make the scale bar on Fig. 2b more visible.

- I suggest to make amplification images on Fig 5b bigger.

- Finally, the title sound a bit large. I would recommend to shorten it. My suggestion is: "Oncostatin M enriched small extracellular vesicles prevents isoproterenol-induced fibrosis and enhances angiogenesis"

Reviewer 3 Report

General comments to Authors

The potential protective effect of extracellular vesicles overexpressing Oncostatin M (OSM) on their surface against fibrosis and cardiac hypertrophy has been assessed in vitro using human cardiac ventricular fibroblasts upon TGF-β1 stimulation and in vivo in mice treated with isoproterenol. The data presented show that, in comparison with control extracellular vesicles, OSM enriched vesicles were able to further reduce proliferation and telo-collagen expression in ventricular fibroblasts, as well to reduce fibrosis, prevent cardiac hypertrophy, and increase angiogenesis in mice treated with isoproterenol.

I think this study is original and interesting,  which may contribute to increase our knowledge on the protective properties of extracellular vesicles.

My only observation concerns the data presentation in the figures. I strongly suggest the authors to improve their quality.

Reviewer 4 Report

The authors have developed modified MSC cell line to produce extracellular vesicles, and tested usefulness of extracellular vesicles to deliver functional proteins in animal experiments. This study is of clinical significance. However, there are several concerns.

1.       Lentiviral vectors are used for gene transfer, but have you verified that no vectors or recombinant genes are present in the secretory vesicles?

2.       In in vivo experiments, extracellular vesicles are administered by intraperitoneal injection, has the transferability of OSM to the heart been quantified?

3.       The discussion section of the manuscript is largely a summary of the results, and a discussion of extracellular vesicles is required.

4.       What does CM on line 83 stand for?